# Association between Active Commuting and Lifestyle Parameters with Mental Health Problems in Chilean Children and Adolescent

**DOI:** 10.3390/bs14070554

**Published:** 2024-06-29

**Authors:** Felipe Caamaño-Navarrete, Indya Del-Cuerpo, Carlos Arriagada-Hernández, Cristian Alvarez, Anelise Reis Gaya, Cézane Priscila Reuter, Pedro Delgado-Floody

**Affiliations:** 1Physical Education Career, Universidad Autónoma de Chile, Temuco 4780000, Chile; marfel77@gmail.com (F.C.-N.); carlosarriagadah@gmail.com (C.A.-H.); 2Department of Physical Education and Sports, Faculty of Sports Sciences, University of Granada, 18010 Granada, Spain; delcuerpo@ugr.es; 3Strength & Conditioning Laboratory, CTS-642 Research Group, Department Physical Education and Sports, Faculty of Sports Sciences, University of Granada, 18010 Granada, Spain; 4Grupo de Investigación Colaborativa para el Desarrollo Escolar (GICDE), Temuco 4780000, Chile; 5Exercise and Rehabilitation Sciences Institute, School of Physical Therapy, Faculty of Rehabilitation Sciences, Universidad Andres Bello, Santiago 7591538, Chile; cristian.alvarez@unab.cl; 6Graduate Program in Human Movement Sciences, Federal University of Rio Grande do Sul, Porto Alegre 90690-200, Brazil; anegaya@gmail.com; 7Graduate Program in Health Promotion, University of Santa Cruz do Sul, Independência Av.2293-Universitário, Santa Cruz do Sul 96815-900, Brazil; cezanereuter@unisc.br; 8Department of Physical Education, Sport and Recreation, Universidad de La Frontera, Temuco 4811230, Chile

**Keywords:** physical activity, childhood, food habits, active transport

## Abstract

Background: Little is known about the association between active commuting (i.e., walking or cycling to school) with lifestyle parameters and mental health in youths. The objective of the present study was to investigate the association between mental health problems and symptoms of depression, anxiety, and stress with lifestyle (i.e., food habits, screen time, physical activity, and sleep quality), active commuting, and gender. Methods: A total of 511 children and adolescents (boys, n = 249; girls, n = 262) aged 10 to 17 years participated in the study. Lifestyle parameters and mental health were evaluated using the Depression Anxiety and Stress Scale (DASS-21). Results: Girls reported higher levels of anxiety (*p* = 0.001), depression (*p* = 0.001), and stress (*p* = 0.001). Mental health problems showed a positive association with gender (girls, β = 3.06, *p* < 0.001) and a negative association with food habits (β = −0.65, *p* = 0.019). Anxiety was positively associated with gender (β = 7.88, *p* < 0.001) and negatively associated with food habits (β = −0.23, *p* = 0.019). Gender (girls) and food habits were also associated with symptoms of depression (β = 2.29, *p* < 0.001 and β = −0.27, *p* = 0.005, respectively). Finally, active commuting was inversely associated with stress (β = −1.24, *p* = 0.008), and stress was positively linked to gender (β = 2.53, *p* < 0.001). Conclusions: Active commuting, lifestyle parameters, and gender were associated with mental health in children and adolescents. Moreover, girls reported higher levels of anxiety, symptoms of depression, and stress.

## 1. Introduction

Childhood and adolescence are critical periods of life characterized by numerous physiological and psychological changes [1]. The development of mental health during these stages is crucial for well-being in later life [2,3], as mental health is widely recognized as a key component of overall well-being [4]. Moreover, good mental health is a state in which an individual develops their abilities, is resistant to the stresses of life, and can make a positive contribution to their peers [5]. In addition, mental health is positively related to social and psychological functioning [6]. Mental health problems during the school years are a global public health issue [7]. Consequently, promoting positive mental health at early stages has been a key focus of health and educational policies in Latin American countries [8]. Despite the importance of maintaining good mental health in early life, it is estimated that a significant percentage of children and adolescents are affected by mental health disorders [9]. Many of these disorders, such as anxiety and depression, often emerge during adolescence [10]. In this sense, previous data indicate an increasing prevalence of mental health disorders among children and young people [11]. Depression is a common mental health problem characterized by low mood, loss of interest or pleasure, low energy, and poor concentration [12].

A growing number of school-aged children are at risk for a myriad of psychological and behavioral problems related to symptoms of depression, which interfere with their interpersonal relationships and impact academic performance [13]. Moreover, anxiety is a term used to describe many anxiety disorders such as panic disorder, social and generalized anxiety, and specific phobias [14]. Indeed, adolescence (characterized by changes in the social environment) may have a negative impact on stress in children and adolescents [10]. Complementing the above, a previous study showed high levels of symptoms of depression, anxiety, and stress in adolescents [15]. These mental health problems are related to poor academic performance [16]. 

Similarly, it has been suggested that there is a positive relationship between a healthy lifestyle and better mental health in children and adolescents [17]. Complementing the above, better food habits (i.e., intake of fruit and vegetable products) are positively associated with mental health in adolescents [18]. Some studies have reported a strong association between better food habits and better psychosocial well-being [19,20], and high-quality food habits are strongly associated with less depressive symptoms [21]. Additionally, there was a recently report in Latin American adolescence that negative feelings are associated with unhealthy lifestyles such as lower nutritional levels, increased screen time, and low physical activity (PA) levels [22]. The current evidence suggests that individuals who show better food habits also show better mental health during the school stage [20]. Likewise, a lack of healthy food habits in children and adolescents may play a critical role in the development of mental health disorders such as depression symptoms [23]. In addition, there is strong evidence for the negative relationship between a high-fat diet and anxiety [24]. There is evidence that high levels of mental health are associated with breakfast quality and total dietary patterns [25]. Hence, educational-health policies should help students develop healthy food and regular PA habits [20]. 

A growing body of literature supports the beneficial effect of PA on mental health in children and adolescents [26]. In this sense, active commuting (i.e., walking or cycling to school) could be a strategy to increase the PA levels in youth [27]. Moreover, there is strong evidence for a positive relationship between active commuting and psychological well-being as well as a healthier body composition [28]. Similarly, previous evidence has shown that subjects with long active commuting times have better mental and physical well-being [27]. In addition, it has been shown that active commuting was related to better, healthier behaviors [29]. Correspondingly, another study found that mental health was poorer when students engaged in less active commuting [30]. Indeed, sex may have an impact on mental health in children and adolescents [31]. In this context, it has been indicated that there are sex differences in adolescents’ mental health problems [32]. A study among adolescents found that compared to boys, girls had worse levels of mental health (i.e., higher rates of depression and severe depression) [33]. However, there is limited information regarding the relationship between active communing, lifestyle parameters, and mental health in Chilean children and adolescents. Therefore, the objective of the present study was to investigate the association between mental health problems and symptoms of depression, anxiety, and stress with lifestyle factors (i.e., food habits, screen time, PA, and sleep quality), active commuting, and gender. We hypothesize that there is an association between mental health problems and stress with lifestyle, active commuting, and sex. 

## 2. Materials and Methods

The present investigation used a quantitative, cross-sectional, and descriptive-associative approach. A total of 511 Chilean students in the age range of 10 to 17 years, from different schools located in the city of Temuco, Chile (south of Chile), participated in this study. The gender distribution was balanced (boys, n = 249; girls, n = 262). Initially, seven schools were invited to participate in the research, of which, five responded favorably. The five selected schools were those willing to participate in the research and provided time for the application of the selected tests. In total, 564 participants answered all the instruments in a computer laboratory. However, it is important to mention that 53 students were excluded during the selection process. The reasons for exclusion included girls who did not meet the criteria established for inclusion in the study or who presented various reasons that affected their participation (n = 30). Similarly, boys who did not meet the inclusion criteria or had other reasons preventing them from participating were identified (n = 23). It should be noted that the sample used in this study was intentional and non-probabilistic, which implies that the selection of participants was carried out specifically and consciously, according to the objectives and criteria that were pre-established by the researchers.

The inclusion criteria adopted for this study were rigorous and covered specific conditions that ensured the integrity of the research. These included the following conditions: (i) the participant belonged to the selected school; (ii) informed consent was obtained from the participants and their parents or guardians, thus guaranteeing the understanding and voluntariness of their participation in the study; (iii) were students to clearly ensure the profile of the target population; and (iv) completed all the data collection instruments. On the other hand, the exclusion criteria were equally detailed and focused on ensuring the suitability of the participants for the evaluations. These were the following: (i) any medical contraindication that prevented an average performance in the evaluations, thus guaranteeing that the results accurately reflected the capabilities of the student population under study and (ii) not being present at the time of the evaluations or not granting informed consent, thus ensuring the validity and ethics of the data collection process. It is important to note that this study strictly adhered to the ethical principles established in the Declaration of Helsinki [2013] which are fundamental to guarantee the protection of the rights of participants in scientific research. Additionally, it received approval from the Ethics Committee of Universidad Autónoma de Chile (ACTA; No. CEC 11-23), which supported the integrity and methodological validity used. The participation of the students in the research was conditional on obtaining their signed consent, as well as the informed consent of their respective parents or guardians.

The research team that evaluated the children and adolescent participants were instructed in the protocols and evaluations prior to data collection. During a morning class, the assessments of mental health, lifestyle, and active commuting were carried out in a computer lab facilitated by the schools. The questionnaires and evaluations were completed individually and in the presence of researchers to resolve any questions. Data collection was performed from April to November 2023 following the Chilean school calendar. 

### 2.1. Main Outcomes

#### Lifestyle

The Krece Plus test was used to evaluate food habits [34]. This test is a very useful tool to predict and prevent nutritional alterations in children and adolescents. It allows the quick and effective detection of any nutritional and physical risks that may lead to obesity. The instrument consists of 16 dichotomous questions that must be answered affirmatively/negatively (yes/no). The items have a score of +1 or −1 according to the established guidelines. The total score was classified according to previous recommendations as follows: (i) from 8 to 12: optimal Mediterranean diet adherence; (ii) from 4 to 7: need to improve eating habits; and (iii) from 0 to 3: very-low-quality diet [34]. The Krece Plus instrument has been used previously in Chilean students [35]. 

In addition, PA per week and screen time (ST) per week were evaluated using the Krece Plus test [36]. The Krece Plus test is a quick questionnaire that classifies lifestyle based on the daily average of hours spent watching television or playing video games per day (ST) and the hours of PA after school per week. This instrument has been previously used in Chilean students to determine their PA per week and ST per day [35]. The duration of sleep was determined through the following question, as was performed in previous studies [37]: “How many hours of sleep do you usually get per day and/or night?” The above question has been used and methodologically accepted in several studies [38,39]. 

### 2.2. Active Commuting

To determine active commuting, students answered the PACO (Pedalea y Anda al COle) questionnaire in a self-reported manner. The PACO investigation is focused on the exploration of travel patterns to and from school among the youth population and was developed by the University of Granada, Spain [40]. Furthermore, this instrument is a tool created to evaluate the active travel habits (walking and/or cycling) of secondary education students to and from school. The PACO questionnaire has been previously used in Chilean students [41]. According to previous recommendations, participants were asked to answer two questions related to the usual mode of commuting to and from school (“How do you go to school?” and “How do you come back from school?” for each day of the week) [42]. The responses were categorized as “active” commuting if the participant walked or cycled and “non-active” commuting if the trip was made by car or bus [41]. The questionnaire has shown a good reliability in a subsample of 219 Chilean schoolchildren [41]. 

### 2.3. Mental Health 

The abbreviated version of the Depression Anxiety Stress Scales (DASS-21) was utilized to identify any mental health problems [43]. This three-dimensional self-report scale was designed to evaluate the presence and intensity of emotional states or symptoms of depression, anxiety, and stress [44,45,46]. The questionnaire comprises 21 questions divided into three subscales: depression (which assesses feelings of dysphoria, hopelessness, devaluation of life, self-hatred, lack of interest, and anhedonia), anxiety (which evaluates experiences of physiological arousal, situational anxiety, and general anxiety), and stress (which evaluates levels of irritability, overwhelming feelings, problems relaxing, and difficulty controlling excessive thinking), with a score from 0 to 3 indicating the degree of intensity concerning the past week (ranging from “It doesn’t describe anything that happened to me or how I felt during the week” to “Yes, this happened to me a lot, or most of the time”). The sum of the answers can range from 0 to 21 points (total score). This instrument has the advantage of being self-reported, brief, easy to administer and answer, and easy to interpret [47]. These questionnaires have been used in Chilean students [48] and have presented adequate psychometric properties [47]. 

### 2.4. Statistical Analysis

The statistical analyses were performed using SPSS^®^ v23.0 software (SPSS^TM^ Inc., Chicago, IL, USA). Normal distribution was tested using the Kolmogorov–Smirnov test. For continuous variables, the values are presented as the mean and standard deviation (SD), whereas for categorical variables, the data are presented as the frequency and percentage (n, %) with the Chi^2^ test results. Differences between genders (boys and girls) were determined using the Student’s T-test. To investigate the association of mental health variables (DASS-21, depression, anxiety, and stress) with lifestyle parameters (food habits, screen time, and PA), active commuting, and gender, a multivariable lineal regression analysis was conducted with the results reported as a beta coefficient (*β*) and their 95% confidence interval (CI). Analyses were conducted using two statistical models; model 1 was not adjusted and model 2 was adjusted by age. *p* value < 0.05 was considered significant. 

## 3. Results

The girls reported higher levels of anxiety, symptoms of depression, and stress than the boys, and presented lower scores for food habits. The boys reported higher screen times and PA. According to sleep time, there were no differences (Table 1).

In Table 2, the frequency of mental health problems according to gender is shown. The girls reported a higher presence of anxiety (extremely severe: men 26.5% vs. women 49.6%, *p* = 0.001), symptoms of depression (extremely severe: men 11.2% vs. women 24.8%, *p* = 0.001), and stress (extremely severe: men 4.4% vs. women 16.0%, *p* = 0.001).

Table 3 shows the association between total mental health according to DASS-21 and anxiety with gender, lifestyle parameters, and active commuting. In the regression analysis, mental health problems showed a positive association with gender (girls, β = 3.06, *p* < 0.001) and an inverse association with food habits (β = −0.65, *p* = 0.019). Anxiety was associated with gender (girls, β = 7.88, *p* < 0.001) and had an inverse association with food habits (β = −0.23, *p* = 0.019).

Gender (girls) and food habits were associated with symptoms of depression (β = 2.29, *p* < 0.001 and β = −0.27, *p* = 0.005, respectively). Finally, active commuting was inversely associated with stress (β = −1.24, *p*= 0.008), and stress was positively linked to gender (girls, β = 2.53, *p* < 0.001) (Table 4).

## 4. Discussion

The objective of the present study was to investigate the association between mental health problems (i.e., total DASS-21), symptoms of depression, anxiety, and stress with lifestyle (i.e., food habits, screen time, PA, and sleep quality), active commuting, and sex. The main findings of this study were as follows: (i) girls reported a higher presence of mental health problems in terms of anxiety, and symptoms of depression and stress; (ii) mental health problems had a positive association with gender (girls) and an inverse association with food habits; (iii) anxiety was linked to gender and had an inverse association with food habits; (iv) gender and food habits were associated with symptoms of depression; and (v) active commuting was inversely associated with stress, and stress was linked to sex. 

The inverse association between mental health, such as anxiety and depression risks, with unhealthy food habits may arise from several of the variables of this research. Recent studies indicated that our food, especially some foods, may be associated with either aggravating or even enhancing the mood while supporting the psychological processes of mental health [21,49]. Moreover, several researchers affirmed that there is proof that eating foods consisting of a variety of vegetables and fruits and avoiding a pro-inflammatory diet, which is composed of mainly of junk food, fast food, and high-meat intake, is not only a gateway to preventing the occurrence of depressive episodes but also clinical depression [50]. Moreover, the evidence suggests that food habits, particularly the intake of an abundance of canned foods, frozen foods, and fast foods, tend to play a vital role in mental health in terms of the depression and anxiety seen in many students [51]. In this context, another study among Spanish adolescents found that better food habits were positively associated with subjective happiness and health-related quality of life (HRQoL) [52]. Indeed, good food habits have been proven to be a positive instrument for improving mental health [53]. In addition, data from a systematic review and meta-analysis showed that there is an inverse relationship between healthy eating habits and the odds of depression [54]. Continuing with the above, it has been indicated that healthy food habits are associated with better perceived stress and well-being [55]. Based on the previously reported findings, healthy food habits during adolescence may be associated with better HRQoL [56]. In addition, the evidence has shown that a pro-inflammatory diet is associated with increases odds of mental health problems in female adolescents [57]. 

The fact that food habits and mental health have a complex interactive relationship was obtained from exhaustive studies [55,58]. Moreover, the influence of particular food items on one’s anxiety and symptoms of depression is known; there is also recent evidence supporting the belief that overall dietary patterns could be related to individuals’ psychological health [59]. Several recent studies claimed that keeping certain dietary habits that are consistent with the Mediterranean diet (that has ample amounts of whole grains, fruits, and vegetables rich in healthy fats) can help to decrease the risk of depressive symptoms and improve certain cognitive functions as well as emotional resilience [60]. In this context, the evidence has shown that adherence to a Mediterranean diet is inversely associated with the odds of experiencing symptoms of depression, anxiety, and stress [61]. The last reason to follow a balanced diet is the gut–brain axis, which connects the gastrointestinal tract with the brain. This two-way communication results in the need to maintain a balanced diet for mental health [62]. A variety of nutrients that are prevalent in plant-based foods, for example, omega-3 fatty acids and vitamin B, are very important in the regulation of neurotransmitter function. Also, they may have anti-inflammatory actions and thus they can protect us from mood disorders [63]. People should attain a good balance of nutrition not only based on food items, but also focus on components such as food–mood connections that may promote good mental and psychological health. Complementary to the above, another study indicated that in females, refined grain dietary patterns and a pro-inflammatory diet was associated with a higher risk of developing depression [64]. On the contrary, the evidence has shown that long-term healthy eating patterns (i.e., anti-inflammatory diet) may prevent mental health problems including symptoms of depression and anxiety [65]. Likewise, another study showed that a Mediterranean diet intervention decreased the symptoms of depression and improved the quality-of-life scores in young males [66].

Active commuting was inversely associated with stress; in contrast, despite its stress-relieving function, which is the result of several factors outlined in the study, the stress response line slope was negative. Certain research has proven that active commuting helps to decrease the level of stress because it provides a chance to be away from the daily routine and devote time to moving [67]. Moreover, directly heading from areas with low air flow and plentiful green spaces may also lead to a decrease in the risk of depression [68]. Moreover, active commuting is thought to be relaxing and pleasing and may possibly relieve stress differently from other methods of transport; cyclists are perceived to be the happiest commuters [69]. The physiological indicators such as blood pressure and galvanic skin response can be used to show that non-motorized travels are less stressful and safer than motorized ones; walking and cycling have been known to better reduce effort and stress [70]. For instance, it has been shown that individuals who cycle or walk to work or school have a heightened sense of well-being and work–life balance compared to those who do not actively commute [71]. Traveling to work by private transport is viewed as highly cumbersome and tedious because of the stress experienced in contrast with other modes of commuting [72] and because it tends to entail the most distinct negative health outcomes among the passive means of transportation. In addition, data from schoolchildren (12,151 females and 9445 males) found that active commuting to school was associated with lower odds of mental health problems [27]. Complementary to the above, it has been indicated that there is a positive association between active commuting and higher well-being in adolescent females [73]. According to the above, another study shown that PA was positively associated with fewer symptoms of anxiety in students [74]. On the contrary, a recent study found that passive commuting was associated with poorer mental health [75]. Other evidence suggests that engaging in actively commuting to school has a positive impact on psychological well-being in school-aged subjects [76].

The girls reported worse mental health than boys (i.e., anxiety, symptoms of depression, and stress), reporting a higher frequency of mental health problems. Females in their adolescence have also reported a higher incidence of emotional stress. This can be attributed to many factors that affect students and hence this outcome was obtained. Studies have proved that girls who are entering into their teens and from the lowest class of society were found to be the most affected psychologically, and thus their psychological well-being was rated the lowest [77,78]. This trend is similar for emotional disturbance, and is also appeared in girls who scored exceptionally high in prosocial behavior, which also protects girls from mental health problems should they arise [78]. Nonetheless, boys are more dominant when it comes to addressing problems and the same situations with absent or threatening parents; therefore, close observations and intervention techniques are required [77,78]. By contrast, in-depth studies have revealed that the high rates of most common mental illnesses are due to women, where reports of social anxiety disorder were much higher [79].

Previous evidence showed that socio-economic status plays a crucial role in mental health outcomes in teenagers and lower economic status often leads to worse mental health [80]. Alongside economic factors, the family setting is also a factor that affects an individual’s mental health [81]. Research has confirmed that individuals who benefit from having relatives that they can lean on and a warm and safe family life have protective factors that can prevent certain mental health problems [82]; it thus becomes clear that a stable and comfortable home environment is the backbone of mental well-being. Additionally, the extent and relationships with peers and the social aspects during adolescence were discovered to be consequential for mental health, meaning that there should be interventions intended to solidify healthy social relationships and emotional support throughout the adolescent population [82].

In the present investigation, the main limitation was the cross-sectional design. Moreover, we used a convenience sample which does not allow the results to be extrapolated. Among the strengths, we could highlight the simplicity of the assessments (which would allow their use and application in healthy lifestyle interventions focused on Chilean students). In the future, we plan to conduct longitudinal studies to better clarify the statistical associations. In addition, we plan to carry out active breaks to see their impact on different variables of interest to schools.

## 5. Conclusions

In conclusion, this study delved into the intricate association between mental health issues like symptoms of depression, anxiety, and stress, and various lifestyle factors among Chilean adolescents. Active commuting, lifestyle parameters, and gender were associated with mental health in children and adolescents. Moreover, there were significant differences in mental health problems between genders, with girls reporting higher levels of anxiety, symptoms of depression, and stress. Notably, food habits were inversely associated to mental health problems, particularly anxiety and symptoms of depression. This highlights the critical role of nutrition in mental health outcomes among adolescents. Additionally, active commuting, such as walking or cycling to school, was associated with lower stress levels, indicating the potential benefits of physical activity on mental well-being. Finally, the present study provides valuable insight into the complex interplay of factors influencing mental health during childhood and adolescence. It emphasizes the importance of promoting healthy lifestyle habits, including nutritious eating and PA, to support better mental well-being among adolescents, with a particular focus on addressing gender-related differences.

## Figures and Tables

**Table 1 behavsci-14-00554-t001:** Comparison of variables according to gender.

	Total (n = 511)	Boys (n = 249)	Girls (n = 262)	*p*-Value _(F-Value)_
Age (y)	13.68 ± 1.64	13.63 ± 1.54	13.72 ± 1.71	*p* = 0.561 _(0.34)_
Lifestyle parameters				
Physical activity (h/day)	1.74 ± 0.96	1.94 ± 0.97	1.55 ± 0.92	*p* < 0.001 _(20.89)_
Screen time (h/day)	3.43 ± 1.50	3.66 ± 1.58	3.23 ± 1.40	*p* < 0.001 _(10.61)_
Sleep time (h)	7.91 ± 1.53	7.98 ± 1.46	7.85 ± 1.60	*p* = 0.327 _(0.96)_
Food habits (score)	4.81 ± 2.69	5.42 ± 2.61	4.23 ± 2.63	*p* < 0.001 _(26.42)_
Active commuting				
No n (%)	302 (50.1%)	137 (55.0%)	165 (63%)	*p* = 0.041
Yes n (%)	209 (40.9%)	112 (45%)	97 (37%)	
Well-being				
Mental health total (score)	25.69 ± 15.17	21.45 ± 13.48	29.73 ± 15.60	*p* < 0.001 _(41.01)_
Anxiety (score)	7.90 ± 5.68	6.31 ± 5.09	9.42 ± 5.80	*p* < 0.001 _(41.37)_
Symptoms of depression (score)	8.09 ± 5.55	6.78 ± 4.99	9.33 ± 5.78	*p* < 0.001 _(28.42)_
Stress (score)	9.71 ± 5.25	8.37 ± 4.93	10.98 ± 5.24	*p* < 0.001 _(33.71)_
HRQoL (score)	23.81 ± 6.84	26.55 ± 6.27	21.21 ± 6.34	*p* < 0.001 _(91.82)_

Data are shown as the mean and ± standard deviation and n (%). A *p* value < 0.05 denotes a significant difference. HRQoL refers to health-related quality of life.

**Table 2 behavsci-14-00554-t002:** Frequency of mental health problems according to gender.

	Total	Boys	Girls	*p* Value
Anxiety				
Absence	146 (28.6%)	94 (37.8%)	52 (19.8%)	*p* = 0.001
Mild	31 (6.1%)	21 (8.4%)	10 (3.8%)	
Moderate	85 (16.6%)	42 (6.9%)	43 (16.4%)	
Severe	53 (10.4%)	26 (0.4%)	27 (10.3%)	
Extremely severe	196 (38.4%)	66 (6.5%)	130 (49.6%)	
Symptoms of Depression				*p* = 0.001
Absence	160 (31.3%)	98 (39.4%)	62 (23.7%)	
Mild	66 (12.9%)	39 (15.7%)	27 (10.3%)	
Moderate	116 (22.7%)	53 (21.3%)	63 (24.0%)	
Severe	76 (14.9%)	31 (12.4%)	45 (17.2%)	
Extremely severe	93 (18.2%)	28 (11.2%)	65 (24.8%)	
Stress				*p* = 0.001
Absence	185 (36.2%)	111 (44.6%)	74 (28.2%)	
Mild	62 (12.1%)	40 (16.1%)	22 (8.4%)	
Moderate	89 (17.4%)	39 (15.7%)	50 (19.1%)	
Severe	122 (23.9%)	48 (19.3%)	74 (28.2%)	
Extremely severe	53 (10.4%)	11 (4.4%)	42 (16.0%)	

Data are shown as n (%). *p* < 0.05 denotes a significant difference.

**Table 3 behavsci-14-00554-t003:** Association between mental health (DASS-21) and anxiety with sex, food habits, and lifestyle parameters.

		β (95%CI)	Beta	SE	t	*p*-Value
Mental health (DASS-21)						
Gender (ref: boys)	Model 1	7.88 (5.18; 10.57)	0.26	1.37	5.75	*p* < 0.001
	Model 2	7.94 (5.26; 10.62)	0.26	1.36	5.82	*p* < 0.001
Food habits (score)	Model 1	−0.65 (−1.15; −0.15)	−0.12	0.25	−2.57	0.011
	Model 2	−0.66 (−1.16; −0.16)	−0.12	0.25	−2.61	0.009
Screen time (h/day)	Model 1	0.38 (−0.48; 1.24)	0.04	0.44	0.87	0.384
	Model 2	0.26 (−0.60; 1.13)	0.03	0.44	0.60	0.547
Sleep time (h)	Model 1	0.16 (−0.68; 1.00)	0.02	0.43	0.38	0.704
	Model 2	−0.05 (−0.90; 0.81)	0.00	0.44	−0.11	0.914
Physical activity (h/week)	Model 1	0.61 (−0.76; 1.99)	0.04	0.70	0.87	0.383
	Model 2	0.69 (−0.68; 2.06)	0.04	0.70	0.99	0.320
Active commuting (ref: no)	Model 1	−2.32 (−4.94; 0.31)	−0.08	1.34	−1.73	0.084
	Model 2	−2.28 (−4.90; 0.33)	−0.07	1.33	−1.72	0.087
Anxiety						
Sex (ref: boys)	Model 1	3.06 (2.05; 4.07)	0.27	(0.51	5.95	*p* < 0.001
	Model 2	3.08 (2.08; 4.09)	0.27	(0.51	6.01	*p* < 0.001
Food habits (score)	Model 1	−0.23 (−0.41; −0.04)	−0.11	(0.10	−2.36	0.019
	Model 2	−0.23 (−0.42; −0.04)	−0.11	(0.10	−2.40	0.017
Screen time (h/day)	Model 1	0.07 (−0.25; 0.40)	0.02	(0.16	0.44	0.662
	Model 2	0.03 (−0.29; 0.35)	0.01	(0.16	0.18	0.855
Sleep time (h)	Model 1	0.17 (−0.15; 0.48)	0.04	(0.16	1.03	0.304
	Model 2	0.09 (−0.23; 0.41)	0.02	(0.16	0.55	0.579
Physical activity (h/week)	Model 1	0.25 (−0.27; 0.76)	0.04	(0.26	0.94	0.348
	Model 2	0.28 (−0.24; 0.79)	0.05	(0.26	1.05	0.292
Active commuting (ref: no)	Model 1	−0.79 (−1.78; 0.19)	−0.07	(0.50	−1.58	0.114
	Model 2	−0.78 (−1.77; 0.20)	−0.07	(0.50	−1.57	0.118

Data are shown as β (95% CI) and *p*-values. *p*-values < 0.05 are considered statistically significant. Model 2 represents β adjusted by age.

**Table 4 behavsci-14-00554-t004:** Association between symptoms of depression and stress with sex, food habits, and lifestyle parameters.

		β (95%CI)	Beta	SE	t	*p*-Value
Depression						
Gender (ref: boys)	Model 1	2.29 (1.28; 3.29)	0.21	0.51	4.48	*p* < 0.001
	Model 2	2.30 (1.30; 3.31)	0.21	0.51	4.52	*p* < 0.001
Food habits (score)	Model 1	−0.27 (−0.45; −0.08)	−0.13	0.09	−2.83	0.005
	Model 2	−0.27 (−0.46; −0.08)	−0.13	0.09	−2.86	0.004
Screen time (h/day)	Model 1	0.17 (−0.15; 0.49)	0.05	0.16	1.02	0.310
	Model 2	0.14 (−0.19; 0.46)	0.04	0.16	0.83	0.409
Sleep time (h)	Model 1	−0.04 (−0.35; 0.28)	−0.01	0.16	−0.23	0.815
	Model 2	−0.09 (−0.41; 0.23)	−0.03	0.16	−0.56	0.573
Physical activity (h/week)	Model 1	0.03 (−0.48; 0.54)	0.01	0.26	0.12	0.902
	Model 2	0.05 (−0.46; 0.57)	0.01	0.26	0.21	0.838
Active commuting (ref: no)	Model 1	−0.29 (−1.26; 0.69)	−0.03	0.50	−0.57	0.566
	Model 2	−0.28 (−1.25; 0.70)	−0.02	0.50	−0.56	0.577
Stress						
Sex (ref: boys)	Model 1	2.53 (1.59; 3.46)	0.24	0.48	5.32	*p* < 0.001
	Model 2	2.55 (1.62; 3.48)	0.25	0.47	5.40	*p* < 0.001
Food habits (score)	Model 1	−0.16 (−0.33; 0.01)	−0.08	0.09	−1.80	0.072
	Model 2	−0.16 (−0.33; 0.01)	−0.08	0.09	−1.85	0.065
Screen time (h/day)	Model 1	0.14 (−0.16; 0.44)	0.04	0.15	0.94	0.346
	Model 2	0.10 (−0.20; 0.40)	0.03	0.15	0.65	0.517
Sleep time (h)	Model 1	0.03 (−0.26; 0.33)	0.01	0.15	0.23	0.817
	Model 2	−0.05 (−0.34; 0.25)	−0.01	0.15	−0.30	0.761
Physical activity (h/week)	Model 1	0.33 (−0.14; 0.81)	0.06	0.24	1.37	0.172
	Model 2	0.36 (−0.11; 0.84)	0.07	0.24	1.51	0.133
Active commuting (ref: no)	Model 1	−1.24 (−2.15; −0.33)	−0.12	0.46	−2.67	0.008
	Model 2	−1.22 (−2.13; −0.32)	−0.12	0.46	−2.66	0.008

Data are shown as β (95% CI) and *p*-values. *p*-values < 0.05 were considered statistically significant. Model 2 represents β adjusted by age.

## Data Availability

The raw data supporting the conclusions of this article will be made available by the authors on request.

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
