# Peer review of "Association between Active Commuting and Lifestyle Parameters with Mental Health Problems in Chilean Children and Adolescent"

_behavsci, 2024, doi:10.3390/bs14070554_

Round 1

Reviewer 1 Report

Comments and Suggestions for Authors

Thanks for the opportunity to review this manuscript on a relevant topic in an under-researched area. All in all I recommend the authors to keep working on this manuscript. My main recommendations are:

-        The quality of the English language needs attention

-        Improve the structure of the Introduction

-        H2: Please explain how children were invited to participate. How many schools, etc? What was the response rate?

-        H2: The inclusion criteria are quite basic and can be referred to as such.

-        Model 2 is adjusted for age, but no data about age is presented. Please consider including more data about the study sample.

-        The goal was to investigate the associations between several variables. Therefore, the association results should be central to the discussion and conclusion.

Comments on the Quality of English Language

The manuscript needs substantial editing.

The introduction can be structured better.

Please ensure consistent use of terminology (e.g. food habits vs diet quality).    

Author Response

Reviewer 1.

Thanks for the opportunity to review this manuscript on a relevant topic in an under-researched area. All in all I recommend the authors to keep working on this manuscript. My main recommendations are:

Response: Dear reviewer, thank you for the opportunity to improve the paper quality, we think that the article has improved a lot.

-        The quality of the English language needs attention

Response:  Thak you for your recommendations we have reviewed the text extensively.

-        Improve the structure of the Introduction.

Response:  We have added your recommendations.

We have added: Childhood and adolescence are critical periods of life characterized by numerous physiological and psychological changes [1]. The development of mental health during these stages is crucial for well-being in later life [2,3], as a mental health is widely recognized as a key component of overall well-being [4]. Moreover, good mental health is a state in which an individual develops their abilities, is resistant to the stresses of life, and can make a positive contribution to their peers [5]. In addition, mental health is positively related to social and psychological functioning [6]. Mental health problems during the school years are a global public health issue [7]. Consequently, promoting positive mental health at early stages has been a key focus of health and educational policies in Latin American countries [8]. Despite the importance of maintaining good mental health in early life, it is estimated that a significant percentage of children and adolescents are affected by mental health disorders [9]. Many of these disorders, such as anxiety and depression, often emerge during adolescence [10]. In this sense, previous data indicate an increasing prevalence of mental health disorders among children and young people [11]. Depression is a common mental health problem characterized by low mood, loss of interest or pleasure, low energy, and poor concentration [12].

-        H2: Please explain how children were invited to participate. How many schools, etc? What was the response rate?

Response:  We have added your recommendations.

We have added:

A total of 511 Chilean students in the age range of 10 to 17 years, from different schools located in the city of Temuco, Chile (south of Chile), participated in this study. Gender distribution was balanced (boys, n=249, girls, n=262). Initially, seven schools were invited to participate in the research, of which five responded favorably. The five selected schools were those willing to participate in the research and provided time for the application of the selected tests. In total, 564 students responded to the selected data collection instruments in a computer laboratory. However, it is important to mention that 53 students were excluded during the selection process.

-        H2: The inclusion criteria are quite basic and can be referred to as such.

Response:  We have added your recommendations.

We have added:

The inclusion criteria adopted for this study were rigorous and covered specific conditions that ensured the integrity of the research. These included the following conditions: i) the participant belonged to the selected school, ii) obtained informed consent from the participants and parents or guardians, thus guaranteeing understanding and voluntariness of their participation in the study and iii) were students, to clearly ensure the profile of the target population and iv) completed to all data collection instruments.

-        Model 2 is adjusted for age, but no data about age is presented. Please consider including more data about the study sample.

      Response:  Dear editor, thank you for your recommendation, in the participants section of material and methods we have that the schoolchildren age range between 10 to 17 years (presented in table 1), therefore, since there is wide difference about age, in the regression model the variable were adjusted by age. Independent of the adjustment, the study variables maintained significant associations. Also considering other recommendations in the process of peer review.

-        The goal was to investigate the associations between several variables. Therefore, the association results should be central to the discussion and conclusion.

Response:  We have changed it according your recommendations.

Reviewer 2 Report

Comments and Suggestions for Authors

First of all, I would like to thank the journal for the trust placed in me to review this article.

This research shows an analysis of how leading a healthy lifestyle may or may not influence the appearance of mental health problems in a sample of Chilean adolescents. I consider it to be a very complete article that needs minor revisions in order to be published:

- It is recommended that none of the keywords be in the title of the article.

- The introduction is very complete with a large number of references, which perfectly cover the object of the study.

- It is recommended to establish a general hypothesis, following the main objective of the research.

- what type of research is carried out? It should be specified at the beginning of the material and method section.

- It would be convenient to include a section on procedure. How were the questionnaires administered: face-to-face or online? were the researchers present to resolve questions? 

- In Table 1, in the “Active Commuting” section, between which factors are the differences in percentages? This aspect is not at all clear.

- Finally, it is recommended to establish a section on limitations and future lines of research.

Author Response

Reviewer 2.

Dear reviewer, thank you for the opportunity to improve the paper quality, we think that the article has improved a lot.

First of all, I would like to thank the journal for the trust placed in me to review this article.

This research shows an analysis of how leading a healthy lifestyle may or may not influence the appearance of mental health problems in a sample of Chilean adolescents. I consider it to be a very complete article that needs minor revisions in order to be published:

- It is recommended that none of the keywords be in the title of the article.

Response:  We have added your recommendations.

We have added:

physical activity; childhood; food habits; active transport.

- The introduction is very complete with a large number of references, which perfectly cover the object of the study.

Response:  Thanks for your comments.

- It is recommended to establish a general hypothesis, following the main objective of the research.

We have added:

We hypothesize that there is an association between mental health problems and stress with lifestyle, active commuting and sex.

- what type of research is carried out? It should be specified at the beginning of the material and method section.

Response:  We have added your recommendations.

We have added:

The present investigation has a quantitative, cross-sectional and descriptive-associative approach.

- It would be convenient to include a section on procedure. How were the questionnaires administered: face-to-face or online? were the researchers present to resolve questions? 

Response:  We have added your recommendations.

We have added:

The research team that evaluated the children and adolescent participants were instructed in the protocols and evaluations prior to data collection. During a morning class day, the assessments of mental health, lifestyle and active commuting were carried out in a computer lab facilitated by the schools. The questionnaires and evaluations were completed individually and in the presence of researchers to resolve any questions. Data collection was performed from April to November 2023 following the Chilean school calendar.

- In Table 1, in the “Active Commuting” section, between which factors are the differences in percentages? This aspect is not at all clear.

Response: dear reviewer, according to the instrument used ACTIVE COMMUTING refers to the active travel habits (walking and/or cycling) of secondary education students to school. According to previous recommendations, participants were asked to answer two questions related to the usual mode of commuting to and from school (How do you go to school and? How do you come back from school? For each day of the week). Responses were categorized as “active” commuting (if the participant walking or cycling) and non-active commuting (the trip was made in a car or bus).

- Finally, it is recommended to establish a section on limitations and future lines of research.

Response:  We have added your recommendations.

We have added:

In the present investigation, the main limitation is the cross-sectional design. Moreover, we used a convenience sample which does not allow the results to be extrapolated.  Among the strengths, we could highlight: (i) the simplicity of the assessments (which would allow their use and application in healthy lifestyle interventions focused on Chilean students). In the future, we plan to conduct longitudinal studies to better clarify statistical associations. In addition, we plan to carry out active breaks to see their impact on different variables of interest to schools.

Round 2

Reviewer 1 Report

Comments and Suggestions for Authors

Compliments to the authors for improving the manuscript. Despite that it has improved significantly, I still have some suggestions.

-        Line 99: I think it unnecessary to specify ‘girls’ here.

-        Line 335: which statistics? Or is this a reference to results in the literature? Please specify.

-        Please check the correctness of the numbers in table 2, column ‘boys’.

Comments on the Quality of English Language

  I still feel some sentences are written oddly. For example, line 109: ‘154 students responded to the selected data collection instruments’. Please review for language usage.  

Author Response

Reviewer 1:

Dear reviewer, thank you for the opportunity to improve the paper quality, we think that the article has improved a lot.

Compliments to the authors for improving the manuscript. Despite that it has improved significantly, I still have some suggestions.

-     Line 99: I think it unnecessary to specify ‘girls’ here.

Response: we have corrected your suggestion.

Therefore, the objective of the present study was to investigate the association between mental health problems, symptoms of depression, anxiety, and stress with lifestyle factors (i.e., food habits, screen time, PA, and sleep quality), active commuting, and gender.

-     Line 335: which statistics? Or is this a reference to results in the literature? Please specify.

Response: we have corrected your suggestion:

Previous evidence showed that socio-economic status plays a crucial role in mental health outcomes in teenagers and is likely when lower economic status often leads to worse mental health [80].

-     Please check the correctness of the numbers in table 2, column ‘boys’.

Response: we have corrected your suggestion.

Table 2. Frequency of mental health problems according to gender.

Total

Boys

Girls

 P value

Anxiety

Absence   

146(28.6%)

94(37.8%)

52(19.8%)

P=0.001

Mild

31(6.1%)

21(8.4%)

10(3.8%)

Moderate

85(16.6%)

42(6.9%)

43(16.4%)

Severe

53(10.4%)

26(0.4%)

27(10.3%)

Extremely severe

196(38.4%)

66(6.5%)

130(49.6%)

Symptoms of Depression

P=0.001

Absence   

160(31.3%)

98(39.4%)

62(23.7%)

Mild

66(12.9%)

39(15.7%)

27(10.3%)

Moderate

116(22.7%)

53(21.3%)

63(24.0%)

Severe

76(14.9%)

31(12.4%)

45(17.2%)

Extremely severe

93(18.2%)

28(11.2%)

65(24.8%)

Stress

P=0.001

Absence   

185(36.2%)

111(44.6%)

74(28.2%)

Mild

62(12.1%)

40(16.1%)

22(8.4%)

Moderate

89(17.4%)

39(15.7%)

50(19.1%)

Severe

122(23.9%)

48(19.3%)

74(28.2%)

Extremely severe

53(10.4%)

11(4.4%)

42(16.0%)

Data are shown as n (%). p < 0.05 denotes significant differences.
